# A Fluorescence Polarization-Based High-Throughput Screen to Identify the First Small-Molecule Modulators of the Human Adenylyltransferase HYPE/FICD

**DOI:** 10.3390/ijms21197128

**Published:** 2020-09-27

**Authors:** Ali Camara, Alyssa George, Evan Hebner, Anika Mahmood, Jashun Paluru, Seema Mattoo

**Affiliations:** 1Department of Biological Sciences, Purdue University, West Lafayette, IN 47907, USA; acamara@purdue.edu (A.C.); george97@purdue.edu (A.G.); ehebner@purdue.edu (E.H.); mahmooa@purdue.edu (A.M.); 2William Henry Harrison High School, West Lafayette, IN 47906, USA; jashpaluru@gmail.com; 3Purdue Institute for Integrative Neuroscience, Purdue University, West Lafayette, IN 47907, USA; 4Purdue Institute for Inflammation, Immunology and Infectious Disease, Purdue University, West Lafayette, IN 47907, USA

**Keywords:** HYPE/FICD, AMPylation/adenylylation, post-translational modification, BiP/GRP78/HSPA5, α-synuclein, fluorescence polarization, high-throughput screening, drug discovery, assay development

## Abstract

The covalent transfer of the AMP portion of ATP onto a target protein—termed adenylylation or AMPylation—by the human Fic protein HYPE/FICD has recently garnered attention as a key regulatory mechanism in endoplasmic reticulum homeostasis, neurodegeneration, and neurogenesis. As a central player in such critical cellular events, high-throughput screening (HTS) efforts targeting HYPE-mediated AMPylation warrant investigation. Herein, we present a dual HTS assay for the simultaneous identification of small-molecule activators and inhibitors of HYPE AMPylation. Employing the fluorescence polarization of an ATP analog fluorophore—Fl-ATP—we developed and optimized an efficient, robust assay that monitors HYPE autoAMPylation and is amenable to automated, high-throughput processing of diverse chemical libraries. Challenging our pilot screen with compounds from the LOPAC, Spectrum, MEGx, and NATx libraries yielded 0.3% and 1% hit rates for HYPE activators and inhibitors, respectively. Further, these hits were assessed for dose-dependency and validated via orthogonal biochemical AMPylation assays. We thus present a high-quality HTS assay suitable for tracking HYPE’s enzymatic activity, and the resultant first small-molecule manipulators of HYPE-promoted autoAMPylation.

## 1. Introduction

Recent years have witnessed a surge in interest of a once-underappreciated post-translational modification (PTM) termed adenylylation (AMPylation) [1,2]. AMPylation entails the breakdown of an adenosine triphosphate (ATP) co-substrate for the covalent transfer of adenosine monophosphate (AMP) to a hydroxyl sidechain-containing residue of a target protein. Among these adenylyltransferases (AMPylases) are the Fic (filamentation induced by cyclic AMP) family of proteins, whose members are present in all domains of life [3]. Fic proteins are characterized by a structurally conserved Fic domain that houses a canonical FIC motif—HXFX(D/E)(G/A)N(G/K)RXXR—where the invariant histidine is required for catalysis [1,2,4]. Early reports on Fic proteins VopS (*Vibrio* outer protein S) and IbpA (immunoglobulin-binding protein A)—from *Vibrio parahaemolyticus* and *Histophilus somni*, respectively—revealed that the AMPylation of small Rho GTPases promotes bacterial pathogenesis [1,2,5,6]. Consequently, previous high-throughput screening (HTS) efforts have aimed to identify small-molecule inhibitors of bacterial Fic-mediated AMPylation [7].

A single Fic protein, called HYPE (Huntingtin yeast-interacting partner E)/FICD, exists in humans [3,8]. HYPE is topologically organized as an N-terminal transmembrane (TM) domain, followed by two tetratricopeptide (TPR) domains, and a C-terminal Fic domain (Figure 1A and Appendix A) [9]. 

The TM domain anchors HYPE to the endoplasmic reticulum (ER) membrane, while the lumenal-facing TPR and Fic domains are involved in substrate binding and catalysis, respectively [2,9,10,11]. Mutation of HYPE’s His363 to Ala (H363A) within its Fic motif renders the enzyme inactive [2,5,9]. Additionally, HYPE AMPylation is regulated by an inhibitory alpha helix, in which a conserved glutamate sterically obstructs catalytically productive ATP binding in the Fic active site [12]. Mutation of Glu 234 to Gly (E234G) relieves this intrinsic auto-inhibition, generating a constitutively active AMPylase [12].

HYPE is a regulator of ER stress via reversible AMPylation of the heat shock chaperone BiP (Binding immunoglobulin Protein) [9,13,14]. Interestingly, HYPE catalyzes both the AMPylation and deAMPylation of BiP, in response to as-yet-unknown physiological signals [15]. During homeostasis, HYPE-directed AMPylation locks BiP in an inactive state marked by reduced co-chaperone (J-protein)-assisted ATPase activity and the resultant decrease in chaperoning misfolded proteins [10,14]. Accumulation of misfolded proteins in the ER triggers deAMPylation of BiP by HYPE, with the concomitant induction of the unfolded protein response (UPR) in an attempt to restore homeostasis [15].

Maintenance of ER homeostasis is a critical aspect of cell fate during several diseases, making the AMPylation state of BiP of paramount importance. Indeed, a prominent molecular strategy toward the treatment of hyper-proliferating cancer cells is activation of the UPR, where BiP has been shown to serve as a sentinel for controlling apoptosis [16]. Likewise, upregulation of the UPR plays a critical role during diabetes and proper folding of insulin [17]. Recent reports also indicate that upregulated BiP can escape the ER to be used as binding and cell entry receptors for SARS-CoV-2 (etiological agent for COVID-19 (corona virus disease 2019)) spike proteins [18,19]. As the cellular stress enacted by cancer triggers the UPR and BiP expression, this underlying disease could synergize with SARS-CoV-2 infection to worsen health outcomes for COVID-19 patients. Whether BiP’s AMPylation status impacts this binding or ER exit awaits further study.

Given its role in regulating protein folding and the UPR via BiP, it comes as no surprise that genetically manipulating the AMPylation status of a HYPE ortholog (Fic-1 in *C. elegans*) leads to enhanced organismal survival under neurodegenerative conditions caused by aggregating proteins such as amyloid-β, α-synuclein (α-syn), or Huntingtin implicated in Alzheimer’s, Parkinson’s, and Huntington’s diseases, respectively [20]. In addition, we have shown that E234G HYPE can directly AMPylate α-syn in vitro and impede disease-linked phenotypes like fibril formation and lipid membrane permeability [21]. A role for another HYPE ortholog (dFic in *D. melanogaster*) has also been established in blindness [22,23].

The canon of HYPE linkage to disease has been further broadened by omics investigations [24,25,26]. A study tracking highly differentially expressed transcripts under glucolipotoxicity in primary beta islets cells showed that the HYPE gene, *FICD*, is both upregulated and methylated [25]. Proteomic research on AMPylated neuronal proteins has also mapped out a role for HYPE in neurogenesis, suggesting that HYPE-promoted AMPylation remodeling is required for proper differentiation of human neurons [26]. Moreover, this same study introduces an ever-expanding portfolio of novel, putative HYPE substrates, the functions of which are being elucidated [26]. Thus, the ability to manipulate HYPE’s enzymatic activity presents an attractive therapeutic target toward the treatment of diseases including cancer, COVID-19, and neurodegeneration.

Despite being positioned at a therapeutic nexus of various disease pathways, HYPE has yet to be targeted for high-throughput drug discovery efforts. Previous work using purified HYPE to immunize alpacas resulted in HYPE-specific nanobodies capable of manipulating AMPylation [27]. However, the therapeutic potential of these macromolecules is restricted by their high cost of production, lack of drug-like qualities, and low biological activity [27]. Thus, there remains an imminent need for the discovery of HYPE-targeted drugs.

Wild-type (WT) HYPE’s adenylyltransferase activity is intrinsically inhibited but, as mentioned earlier, can be de-repressed via an E234G mutation [2,5,9,12,27]. Exploiting these enzymatically distinct states, we thus set out to discover small-molecule activators and inhibitors of WT and E234G HYPE, respectively. To this end, we developed a dual screen based on the fluorescence polarization (FP) of an ATP analog, N6-(6-amino)hexyl-ATP-5-carboxyl-fluorescein (Fl-ATP) (Figure 1B) [7,28]. When unattached to WT HYPE or compound-inhibited E234G HYPE, Fl-ATP undergoes rapid rotation to depolarize plane-polarized light, and a basal FP signal is generated. Attachment of Fl-AMP via autoAMPylation (E234G HYPE or compound-activated WT HYPE) allows light to remain polarized for a high-FP signal (Figure 1C). We challenged this assay with four diverse chemical libraries (covering nearly 10,000 compounds for each screen) to discover the first small-molecule modulators of HYPE-mediated autoAMPylation. These hits were then assessed for their dose dependencies and validated via orthogonal biochemical counterscreens.

## 2. Results and Discussion

### 2.1. Assay Design

WT HYPE exhibits basal levels of AMPylation in vitro [2,5,9,27], while an E234G mutation results in a robust, constitutively active adenylyltransferase [12]. Therefore, we selected WT HYPE as our negative control for AMPylation and E234G HYPE as our positive control. To overcome the challenge posed by HYPE’s hydrophobic N-terminus in obtaining soluble recombinantly purified full-length protein, we purified two truncated forms of HYPE: (1) Δ45 HYPE, which encodes aa 46–456 of HYPE, thus deleting its TM domain, and has previously been characterized extensively [9]; and (2) Δ102 HYPE, which can be purified to higher yields, and corresponds to what was used previously for solving the crystal structures of HYPE [29,30]. To note, Δ102 HYPE eliminates two of the four identified sites of HYPE autoAMPylation [9,24]. However, this deletion (Figure 1A) did not result in a discernable reduction in autoAMPylation signal relative to the longer Δ45 HYPE (Appendix A).

Using Fl-ATP as a nucleotide source, we monitored changes in autoAMPylation between the previously described HYPE constructs. When free in solution, the relatively small Fl-ATP is subject to rapid Brownian motion, enabling it to depolarize plane-polarized light. However, in the event of covalent attachment to a large protein (i.e., AMPylation), Fl-AMP undergoes slow rotation in which the plane-polarized light remains polarized. This latter condition allows for detection and quantification of AMPylation events. Further, the readout is homogenous and amenable to HTS processing using a standard microplate format and automated liquid handling [7,28].

### 2.2. Assay Development and Optimization

We next tested our FP assay’s ability to distinguish between the AMPylation levels of hypo-active WT and hyper-active E234G HYPE. To ensure we were detecting authentic AMPylation and not the stable, noncovalent binding of Fl-ATP to HYPE, we also assessed the FP signal of catalytically inactive E234G/H363A HYPE. Enzymatically dead mutants are known to serve as in vitro substrate traps, a property used in the past for pulling down HYPE’s interacting partner, BiP [9,31]. We reasoned that if the stable binding of Fl-ATP to inactive HYPE occurs at HYPE’s ATP binding pocket within the Fic domain, a false positive signal would be detected. Additionally, we incubated our enzymes with known in vitro substrates α-syn and BiP [9,21]. As expected, we saw a significantly higher FP signal from E234G HYPE than WT or E234G/H363A (Figure 1D), indicating a lack of adenylyltransferase activity or stable Fl-ATP binding. Of note, a ten times molar excess of substrate failed to significantly improve total AMPylation over auto-AMPylation (Figure 1D). This finding is likely due to the higher proximity-dependent availability of auto-AMPylation sites relative to those of substrates.

To assess conditions under which optimal AMPylation would occur, we compared two well-established buffers routinely used in AMPylation reactions [9,15,21,30,32]. These included a minimal buffer containing 50 mM HEPES (pH 7.5) and 1 mM MnCl_2_; and a complete buffer with 50 mM HEPES (pH 7.4), 150 mM KCl, 1 mM CaCl_2_, 10 mM MgCl_2_, and 0.1% Triton-X 100. AMPylation signals remained consistently high for E234G HYPE and low for WT HYPE, irrespective of buffer choice (Figure 2A). There did appear to be an enhancement in WT HYPE AMPylation under complete buffer conditions, but these levels were still basal relative to E234G. Of the various buffer components, only Mn^2+^, Ca^2+^, and Mg^2+^ are reported to be indispensable for AMPylase activity [9], owing to their ability to coordinate ATP’s α and β phosphates in the Fic catalytic pocket [29,30]. Our previous findings suggested a hierarchy of AMPylation efficiencies among divalent cations, with Mn^2+^ being the preferred metal, followed by Mg^2+^ [9]. We therefore conducted a direct comparison between Mn^2+^ and Mg^2+^ buffers using our FP assay. This resulted in no significant difference in E234G HYPE AMPylation, while WT HYPE levels remained basal (Figure 2B).

DMSO (dimethyl sulfoxide) is the standard vehicle for solvating the polar and nonpolar compounds found in chemical libraries. Despite widespread usage, DMSO is not without its drawbacks, and has been documented to interfere with the enzymatic activities of multiple proteins [33]. In anticipation of our HTS campaign, we added DMSO to minimal and complete buffers to assess its impact on AMPylation. Under both buffer conditions, the AMPylation signal for E234G HYPE decreased significantly (Figure 2C,D), an observation consistent with previous reports of DMSO-mediated enzyme inhibition [33].

To ensure we were catching the reaction in its linear range, we performed a time-course AMPylation experiment using constant concentrations of E234G HYPE and Fl-ATP. These data suggested that a 10 min timepoint would give a suitable signal (Figure 3A,B) while allowing sufficient time to process successive plates by automated liquid handlers during HTS. We next ran reactions at various concentrations of Fl-ATP to determine the signal detection threshold. This step was especially critical considering the expansive sizes of chemical libraries, and the relatively high cost of Fl-ATP (compared to other reaction components). Further, we wanted to select a sub-K_m_ concentration of Fl-ATP to avoid biasing our assay against compounds that are (Fl-)ATP-competitive. Accordingly, we chose an Fl-ATP concentration of 25 nM to conduct the screening assay (Figure 3C).

Before challenging compound libraries with our assay, we first needed to evaluate its robustness and scalability. For this, we turned to an assay quality analysis [34], in which 384-well plates were loaded with either WT or E234G HYPE for 10 min in minimal AMPylation buffer. At our standard enzyme concentration of 200 nM, we observed excellent Z’ (i.e., 1 > Z’ ≥ 0.5) and S/B (signal/background) values: 0.75 and 3.87, respectively (Figure 4A). Doubling the enzyme concentration to 400 nM resulted in further extension of our assay window (Figure 4B).

To further evaluate the validity of our assay, we reasoned that if Fl-ATP truly mimics ATP in AMPylation mediated by E234G-HYPE, then addition of unlabeled ATP would collapse the E234G HYPE-associated FP signal to that of WT HYPE. We tested this assumption by increasingly titrating unlabeled ATP into our standard reaction. Indeed, we saw a concentration-dependent inhibitory effect (Appendix A), indicating the ability of our assay to detect bona fide chemical modulators of AMPylation.

### 2.3. WT HYPE Activator High-Throughput Screen

We used our optimized AMPylation assay to screen four diverse chemical libraries—LOPAC^1280^ (Library of Pharmacologically Active Compounds), Spectrum, MEGx, and NATx—totaling 9680 compounds (Table 1). 

In the first portion of our dual screen, we challenged these libraries against WT HYPE toward the discovery of AMPylation activators. These screens yielded an initial overall hit rate of 0.44% at an empirically determined cut-off of 20% activation relative to controls (Figure 5A). Moreover, the screen maintained suitable Z’ and S/B values throughout, with minimal plate-to-plate variability (Figure 5B) From these initial hits, we then manually eliminated any compounds possessing outlier parallel intensities, to yield a final HTS hit rate of 0.32%. This measure was taken to control for false positive compounds capable of auto-fluorescing or quenching fluorescence emanating from Fl-ATP. While our 20% activation threshold was empirically determined to produce a manageable number of hits, it did conform to standard assay window requirements (i.e., greater than three standard deviations above the negative control mean) [34]. As expected, libraries enriched in pharmacologically active compounds (LOPAC and Spectrum) displayed more hits relative to the less bioactive natural products from the MEGx library (Appendix A). Strikingly, while most of the synthesized compounds in NATx failed to activate AMPylation, several hits stemming from the same source plate were clustered together. Indeed, upon visual inspection of these NATx hit structures, there was a clear emergence of several common functional groups (Figure 7F), suggestive of a similar mechanism of action.

### 2.4. WT HYPE Activator Microplate Validation

We next validated our cherry-picked activator hits in an independent replication of the same HTS assay. Many of our previously discovered hits failed this second-pass selection, having an R^2^ value of 0.17 (Figure 6A). This could be attributed to a relatively permissive 20% hit definition, as selection of a more stringent criterion of 50% activation more than doubled the R^2^ value (Appendix A). Of note, grouping hits based on their libraries of origin revealed a high correlation among LOPAC and Spectrum, while the NATx compounds showed low confirmation (Figure 6B–E). This finding is consistent with the notion that bioactives are likely to have high, reproducible activities when compared to novel, synthetic molecules [35].

Using biological activity, chemical diversity, and commercial availability as selection criteria, we advanced six activators (**A1**–**A6**) for follow-up validation (Figure 5A, Figure 6A and Figure 7A–F, Table 2). 

Concentration–response curves were generated for these hits using minimal buffer with or without 0.1% Triton X-100 detergent, to control for false positive FP signals arising from nonspecific aggregation. In the absence of detergent, we observed robust dose-dependencies among all activators, with low micromolar EC_50_ values (Figure 8A–E; **A6** data not available). With the addition of detergent, however, the putative activators saw a drastic decrease in signal and were only active at high micromolar concentrations (Appendix A). In fact, **A3** reported no dose-dependency at all, even at concentrations up to 1000 μM (Appendix A). Given their long alkyl chains (Figure 7, compounds **A2**–**A5**), these compounds may have been aiding in light polarization via aggregation-assisted trapping of Fl-ATP, rather than true AMPylation [36].

### 2.5. WT HYPE Activator In-Gel Fluorescence Counterscreen

To further validate these hits, we turned to an orthogonal biochemical assay. In addition to detecting FP, the Fl-ATP fluorophore is amenable to in-gel fluorescence AMPylation assays [7,28]. Unlike the FP-based assay—in which AMPylation is indirectly inferred via reduced rotation of the fluorophore—in-gel assays track the direct modification of proteins separated by SDS-PAGE (sodium dodecyl sulfate polyacrylamide gel electrophoresis). Using our typical Δ102 construct, we confirmed two of the putative activators (**A1** and **A4**) (Figure 9A). In the presence of detergent, however, **A4** was unable to induce AMPylation above the DMSO control (Figure 9B). As seen in the concentration–response curve (Appendix A), addition of detergent is likely reversing the aggregative effect of **A4**.

Crystallographic analysis of WT and E234G HYPE indicates a compact, primarily alpha-helical structure [29,30]. These studies, however, were performed with Δ102 constructs; no structures of Δ45 HYPE have been published. We therefore sought to confirm the efficacy of our hits with the larger, more physiologically relevant construct. To this end, we repeated the previous in-gel AMPylation assay with Δ45 HYPE in the presence and absence of Triton X-100. Without detergent, **A1** was the only active compound (Figure 9C); however, the inclusion of detergent led to a significant increase in bioactivity of **A1** and **A4**–**A6** (Figure 9D). With all of their Hill’s coefficients being greater than one, the emergence of bioactivity in the larger Δ45 HYPE could result from allosteric cooperativity (Appendix A) [37].

Of all the putative WT HYPE activators, **A1** (calmidazolium; Figure 7) displayed the strongest FP signal and the most consistent in-gel fluorescence AMPylation signal. Strikingly, this very compound was discovered in a previous HTS campaign as an inhibitor of the bacterial Fic adenylyltransferase VopS [7]. Here, calmidazolium had mechanisms of inhibition noncompetitive to ATP and competitive to its substrate Cdc42 [7]. Given the high evolutionary conservation among Fic domain-containing proteins, we hypothesized a broader interaction between calmidazolium and the Fic domain writ large. Accordingly, we challenged another bacterial Fic AMPylase, IbpA-Fic2, with our **A1** activator. Like VopS, IbpA-Fic2 preferentially acts on a Q61L mutant of the human GTPase Cdc42 during bacterial pathogenesis [1,2,6]. We thus predicted calmidazolium to outcompete Cdc42 for IbpA-Fic2 and perturb AMPylation. Alternatively, calmidazolium could enhance IbpA-Fic2 AMPylation in a manner resembling its interaction with HYPE. However, we detected no manipulation of AMPylation by any of the tested compounds relative to the DMSO control (Appendix A). The activity of calmidazolium against HYPE and VopS is likely attributable to its vast biological promiscuity, rather than some Fic-specific mechanism [38,39,40].

### 2.6. E234G HYPE Inhibitor High-Throughput Screen

Holding all reaction parameters static from our WT HYPE activator HTS, we screened the same four libraries (Table 1) for inhibition of Δ102 E234G HYPE. We saw a high overall inhibition rate of 0.98% at a 50% inhibition cut-off, even after adjustment for auto-fluorescent and fluorescent-quenching molecules (Figure 10A). We therefore chose a more stringent 50% threshold to narrow our hits down to a more manageable level. The greater rate of inhibitor hits relative to activators is as expected, given the ease with which enzymatic activity can be disrupted versus enhanced [41]. Consistent with the activator screen, the LOPAC, Spectrum, and NATx libraries yielded more hits than the synthetic compounds from MEGx (Figure 10A). To assess compound activity reproducibility, we randomly selected a plate from the NATx library. This plate showed strong, positive correlation (Appendix A), thus demonstrating the high reproducibility of our assay. Further, we report suitable Z’ and S/B values throughout the screen (Figure 10B).

### 2.7. E234G HYPE Inhibitor Microplate Validation

Using an independent replication of the same HTS assay, we next validated our E234G HYPE inhibitor hits. These hits saw much greater correlation than the activators—as determined by a higher R^2^ value—likely due to their more stringent hit definition of 50% inhibition (Figure 11A). These correlations were generally consistent across libraries (Figure 11B–E). Additionally, correlation among LOPAC compounds was enriched relative to other libraries, possibly resulting from their exclusively bioactive identity (Table 1).

In accordance with the findings of our unlabeled ATP assay (Appendix A), we discovered multiple chemical structures analogous to ATP (data not shown). Indeed, one of our top inhibitors, the ATP analog diadenosine tetraphosphate (Ap4A), was previously identified in a screen targeting the bacterial Fic effector protein VopS [7]. Of note, E234G HYPE-mediated AMPylation of its putative substrate histone H3 was shown to be inhibited by Ap4A in vitro [7], presumably via an ATP-competitive mode of action.

Employing similar selection criteria as with the activators, we isolated eight inhibitors (**I1**–**I8**) for follow-up investigation (Figure 10A Figure 11A and Figure 12A–H, Table 3). 

For this, we performed concentration–response experiments in the presence of detergent. Under these conditions, only three of the compounds (**I1**–**I3**) displayed dose-dependency (Figure 13A). After fitting the concentration–response data of **I1**–**I3** to the IC_50_ equation (Equation (6), see Methods), we obtained a low micromolar inhibitory potency for **I2**, while **I1** and **I3** were in the mid-micromolar range (Figure 13B–D). To assess whether the reduced bioactivity of **I1** and **I3** is a function of aggregative forces, we repeated this experiment in the absence of detergent. Unlike in the activator screen—where compound aggregation can sequester Fl-ATP to produce a false positive signal—aggregation of putative inhibitors poses a different threat. Here, aggregative compounds can form unwanted colloidal complexes with the enzyme, thus restricting catalysis [36,42]. Indeed, **I1**′s IC_50_ value dropped substantially in detergent-free solution; the IC_50_ value for **I2** remained in the low micromolar range (Appendix A). The activity of **I3**, however, saw no improvement over its initial mid-micromolar range, indicating an impartial response to detergent.

### 2.8. E234G HYPE Inhibitor In-Gel Fluorescence Counterscreen

To confirm their inhibitory potencies via an assay that measures direct protein AMPylation, we subjected **I1**–**I8** to our in-gel fluorescence counterscreen. Further, we wanted to assess the impact of detergent on their activities. As predicted, of our five initially confirmed inhibitors, only two of them (**I2** and **I5**) survived detergent treatment (Figure 14A,B). This trend was largely reproduced using the Δ45 HYPE construct, with **I2** retaining consistently high efficacy (Figure 14C,D).

Using molecular docking of **I2** to a solved crystal structure of HYPE, we sought to gain insight into its preferred orientation and binding affinity. Indeed, we find that **I2** consistently binds to the Fic active site in silico with favorable energetics in the top five searches (Appendix A). Our biochemical data were further corroborated by the weak binding affinity and inconsistent docking of our negative control, **I8** (Appendix A). Given its reproducible bioactivity across assays, assay conditions, and target variants, **I2** (aurintricarboxylic acid) holds promise as both a therapeutic intervention and as a research tool for manipulating AMPylation in vitro. Moreover, aurintricarboxylic acid satisfies Lipinski’s rule of five, indicating its drug-likeness [43]. However, much like our top activator (calmidazolium), aurintricarboxylic acid suffers from broad target promiscuity [44,45,46]. The potential of aurintricarboxylic acid as a HYPE-specific therapeutic agent, therefore, depends on subsequent hit-to-lead optimization.

Using our top three inhibitors (**I1**–**I3**), we next assessed whether this compound-mediated disruption of AMPylation could be a pan-Fic phenomenon. For this, we turned to our bacterial Fic IbpA-Fic-2 and its substrate Q61L Cdc42. Interestingly, the HYPE inhibitors failed to inhibit AMPylation of Q61L Cdc42 by IbpA-Fic2 (Appendix A). Given the high degree of homology between the Fic domains of HYPE and IbpA-Fic2, our data suggest these inhibitors are HYPE-specific and do not target a common Fic active site.

## 3. Materials and Methods

### 3.1. Protein Expression and Purification

Recombinant His_6_-SUMO-tagged proteins were purified as previously described [21]. Specifically, HYPE constructs (WT, E234G, and E234G/H363A) and WT BiP proteins were cloned into pSMT3 plasmids and expressed in *E. coli* BL21-DE3-RILP (Stratagene) in LB medium containing 50 μg/mL of kanamycin to an optical density of A_600_ = 0.6. Protein expression was induced with 0.4 mM IPTG for 12–16 h at 18 °C. Lysis was performed on frozen pellets dissolved in lysis buffer (50 mM Tris, 250 mM NaCl, 5 mM imidazole, 1 mM PMSF, pH 7.5). Lysed cells were centrifuged at 15,000× *g* for 50 min. Supernatants were poured over cobalt resin. Resin was washed with wash buffer (50 mM Tris, 250 mM NaCl, 15 mM imidazole, pH 7.5). Tagged proteins were eluted with elution buffer (50 mM Tris, 250 mM NaCl, 350 mM imidazole, pH 7.5). The His_6_-SUMO tags were cleaved by incubating proteins with ULP1 overnight at 4 °C. The protein mixture was diluted in wash buffer without imidazole and re-applied into a cobalt column. The flow-through containing cleaved protein was further purified by size-exclusion chromatography in a buffer containing 100 mM Tris and 100 mM NaCl, pH 7.5. Fractions containing HYPE were verified for purity by SDS-PAGE and pooled together. Protein concentrations were measured spectrophotometrically at A_280_. Proteins were flash-frozen and stored at −80 °C in storage buffer [50 mM Tris, 300 mM NaCl, 10% (*v*/*v*) glycerol, pH 7.5].

His_6_-tagged WT α-syn was expressed and purified similarly as described above with the following modifications. WT α-syn was cloned into a pT7-7 plasmid and expressed in *E. coli* BL21-DE3-RILP (Stratagene) in LB medium containing 100 μg/mL of ampicillin to an optical density of A_600_ = 0.6. Protein expression was induced with 0.4 mM IPTG for 3–4 h at 30 °C. Lysis was performed on frozen pellets dissolved in lysis buffer. Lysed cells were centrifuged at 15,000× *g* for 50 min. Supernatants were poured over cobalt resin. Resin was washed with wash buffer. Tagged proteins were eluted with elution buffer. Eluted protein was further purified by size-exclusion chromatography in a buffer containing 100 mM Tris and 100 mM NaCl, pH 7.5. Fractions containing WT α-syn were verified for purity by SDS-PAGE and pooled together. Protein concentrations were measured spectrophotometrically at A_280_. Proteins were flash-frozen and stored at −80 °C in storage buffer.

GST-tagged WT IbpA-Fic2 and Q61L Cdc42 were bacterially expressed and purified as previously described [5].

### 3.2. Assay Development

#### 3.2.1. In-Gel Kinetics

For the E234G HYPE time-course experiment, 0.2 μM of Δ102 HYPE enzyme was incubated with 25 nM Fl-ATP in minimal AMPylation buffer (50 mM HEPES, 1 mM MnCl_2_, pH 7.5) at 37 °C in the dark for the specified times. Reactions were stopped with 4X SDS loading dye, boiled for 5 min at 95 °C, and run on 12% SDS-PAGE gels. Gels were imaged on a Typhoon 9500 FLA imager (General Electric, Chicago, IL, USA) for fluorescence at 473 nm. Imaged gels were then stained with Coomassie Brilliant Blue, destained, and imaged for protein concentration. The Fl-ATP concentration experiment was performed as previously described, but with specified concentrations of Fl-ATP, and at a 10 min timepoint.

#### 3.2.2. Microplate Optimization

Buffer comparison experiments were performed by incubating 0.2 μM of Δ102 HYPE enzyme with 25 nM Fl-ATP (final concentrations) in minimal or complete (50 mM HEPES (pH 7.4), 150 mM KCl, 1 mM CaCl_2_, 10 mM MgCl_2_, 0.1% Triton-X 100) AMPylation buffer in the dark at room temperature for 10 min in 384-well black-bottom, black-walled microplates (20 μL reaction volume). Where specified, buffers were supplemented with 1% DMSO. Microplates were snap-centrifuged at 1000× *g* for 10 s. Microplates were loaded onto a BioTek BioStack NEO2 (Agilent, Winooski, VT, USA) plate-reader and assessed for fluorescence polarization with 485/530 nm filters and a 55/50 gain adjustment. Where specified, enzymes were incubated with a 10× molar excess of substrate.

A Multidrop 384 reagent dispenser (ThermoFisher, Waltham, MA, USA) was used to add 0.2 or 0.4 μM Δ102 HYPE, then 25 nM Fl-ATP (final concentrations) to 384-well black/black microplates (20 μL reaction volume). Both enzyme and nucleotide were dissolved in minimal AMPylation buffer. AMPylation reactions were incubated for 10 min at room temperature in the dark. Microplates were snap-centrifuged at 1000× *g* for 10 s, then transferred to a BioTek BioStack NEO2 plate-reader. Reactions were assessed for fluorescence polarization with 485/530 nm filters and a 55/50 gain adjustment. Z’ and S/B (signal/background) values were determined by fitting the data to Equations (1) and (2), respectively,
(1)$$Z\prime =1-\frac{3\left(\sigma_{p}\;\;+\;\sigma_{n}\right)}{\left|\mu_{p}\;-\;\mu_{n}\right|}$$
(2)$$S/B=\frac{\mu_{p}}{\mu_{n}}$$
where *µ_p_* and *µ_n_* are the means of the positive (E234G HYPE) and negative (WT HYPE) controls, respectively; and *σ_p_* and *σ_n_* are the standard deviations of the positive and negative controls, respectively.

### 3.3. High-Throughput Screen

An Echo Liquid Handler (Labcyte, San Jose, CA, USA) was used to transfer 200 nL of DMSO-dissolved compounds via acoustic-coupled ejection from 10 mM source plates into 384-well black, flat-bottom, black-walled microplates, for a final compound concentration of 10 μM. Compounds were sourced from the following libraries: LOPAC^1280^ (Sigma Aldrich), The Spectrum Collection (Microsource Discovery Systems), MEGx (Analyticon Discovery), and NATx (Analyticon Discovery). The compounds went into columns 3–22 of each plate, while columns 1, 2, 23, and 24 were reserved for an equivalent volume of 100% DMSO controls.

A Multidrop 384 reagent dispenser was used to pipette 0.4 μM of Δ102 WT HYPE (final concentration) dissolved in minimal AMPylation buffer into columns 1–22 of the activator plates, and columns 1 and 2 of the inhibitor plates as positive controls. Then, 0.4 μM of Δ102 E234G HYPE was similarly added to columns 3–24 of the inhibitor plates, and columns 23 and 24 of the inhibitor plates as positive controls. Enzymes were then incubated with either compounds or DMSO for 10 min at room temperature.

A Multidrop 384 reagent dispenser was used to pipette 25 nM of Fl-ATP (final concentration) into all wells of each plate, giving a final reaction volume of 20 μL. Plates were then incubated for 10 min at room temperature in the dark, followed by a snap-centrifugation at 1000× *g* for 10 s. Microplates were loaded onto a BioTek BioStack stacker and read in succession for fluorescence polarization with 485/530 nm filters and a 55/50 gain adjustment. Fluorescence polarization values were converted to fractional activation (FA) or fractional inhibition (FI) values according to Equations (3) or (4), respectively:(3)$$\mathrm{FA}=\frac{\left(x\;-\;\mu_{n}\right)}{\mu_{n}}$$
(4)$$\mathrm{FI}=\frac{\left(X\;-\;\mu_{p}\right)}{\mu_{p}}$$
where *µ_p_* and *µ_n_* are the means of the positive (E234G HYPE) and negative (WT HYPE) controls, respectively; and *x* is the measured value of fluorescence polarization.

Microplate “cherry-picking” validations were performed as described above.

### 3.4. Concentration–Response Curves

A Multidrop 384 reagent dispenser (ThermoFisher, Waltham, USA) was used to pipette 0.4 μM of Δ102 WT or E234G HYPE (final concentrations) dissolved in minimal AMPylation buffer with 0.1% Triton X-100 (final concentration) into designated microplate wells. In addition, 10 μM of DMSO-dissolved compounds (final concentrations) or an equivalent volume of 100% DMSO was manually pipetted into their specified wells. Enzymes were incubated with compounds or DMSO for 10 min at room temperature. The reagent dispenser was then used to pipette 25 nM of Fl-ATP (final concentration) into each well, followed by a 10 min incubation at room temperature in the dark. Plates were snap-centrifuged at 1000× *g* for 10 s, then transferred to the BioTek BioStack NEO2 plate-reader. Reactions were assessed for fluorescence polarization with 485/530 nm filters and a 55/50 gain adjustment. EC_50_ and IC_50_ values were determined by fitting polarization values to Equations (5) and (6), respectively,
(5)$$Y=\frac{\left(Y_{max\;}\;X\;{\left[I\right]}^{n}\right)}{\left({\left[{\mathrm{EC}}_{50}\right]}^{n}\;+{\left[I\right]}^{n}\right)}+Y_{min}$$
(6)$$Y=\frac{\left(Y_{max\;}\;X\;{\left[I\right]}^{n}\right)}{\left({\left[{\mathrm{IC}}_{50}\right]}^{n}\;+{\left[I\right]}^{n}\right)}$$
where *Y_max_* is the maximum polarization signal; *I* is the μM concentration of activator or inhibitor; *n* is the Hill slope; EC_50_ or IC_50_ are the inflection point concentrations; and *Y_min_* is the baseline polarization response.

### 3.5. Fluorescence in-Gel Counterscreen

Here, 1 μM of Δ102 or Δ45 WT HYPE (final concentration) were preincubated with 10 μM of DMSO-dissolved compound (final concentration) or an equivalent volume of 100% DMSO at room temperature for 10 min. Reactions were run for 1 h at 37 °C in the dark, and began with the addition of 1 μM Fl-ATP (final concentration). Alternatively, 0.4 μM of Δ45 or Δ102 E234G HYPE were preincubated with 10 μM of DMSO-dissolved compound (final concentration) or an equivalent volume of 100% DMSO at room temperature for 10 min. Reactions were run for 1 h at 37 °C in the dark, and began with the addition of 25 nM Fl-ATP (final concentration). All reactions were performed in minimal AMPylation buffer (50 mM HEPES, 1 mM MnCl_2_, pH 7.5) with or without 0.1% Triton X-100 (final concentration) as specified. All reactions were quenched with 4X SDS loading dye and boiled for 5 min at 95 °C, and the samples were run on 12% SDS-PAGE gels. Gels were imaged on a Typhoon 9500 FLA imager (General Electric) for fluorescence at 473 nm. Imaged gels were then stained with Coomassie Brilliant Blue, destained, and imaged for protein concentration.

All bacterial Fic specificity assays were conducted with 1 μM GST-tagged IbpA-Fic2, 10 μM GST-tagged Q61L Cdc42, and 25 nM Fl-ATP. All other reaction parameters were kept as per HYPE AMPylation assays.

### 3.6. Molecular Docking

Molecular docking to predict protein-compound co-structure and ∆G was done with the SwissDock (http://www.swissdock.ch/) web server. Following program requirements, a protein structure and ZINC (ZINC is not commercial) database accession number were provided [47]. Apo WT HYPE (amino acids 103–445; PDB: 4u04 chain A) was selected due to its free active site and similar overall structure to E234G HYPE. Resultant co-structures were sorted by the top five docking clusters, and the most optimal predicted ∆G values were imaged with the UCSF (University of California San Francisco) Chimera program [48].

## 4. Conclusions

Despite residing at the regulatory intersection of several disease pathways, HYPE has yet to be targeted for an HTS campaign. Here, we present a novel, robust, dual screening assay capable of monitoring changes in HYPE autoAMPylation in vitro. Further, this assay has been optimized for high-throughput scale and demonstrates reproducibility in identifying target hits. In conjunction with our post-HTS counterscreens, we have established a reliable pipeline for hit validation. Indeed, we have discovered at least two promising modulators of HYPE’s adenylyltransferase activity—calmidazolium (**A1**) and aurintricarboxylic acid (**I2**). Though their activities against multiple known human targets restrict usage for therapeutic intervention, these compounds nevertheless remain promising candidates for hit-to-lead optimizations. Moreover, the use of these hits as research tools will aid in the mechanistic elucidation of HYPE’s cellular role in the UPR and other disease-associated pathways. Ongoing follow-up studies on remaining HTS hits not initially counterscreened are sure to provide additional activators and inhibitors of HYPE-mediated AMPylation. The high-quality assay described herein is also fully scalable for HTS processing and lends itself to the screening of larger, more diverse chemical libraries with more drug-like potential.

## Figures and Tables

**Figure 1 ijms-21-07128-f001:**
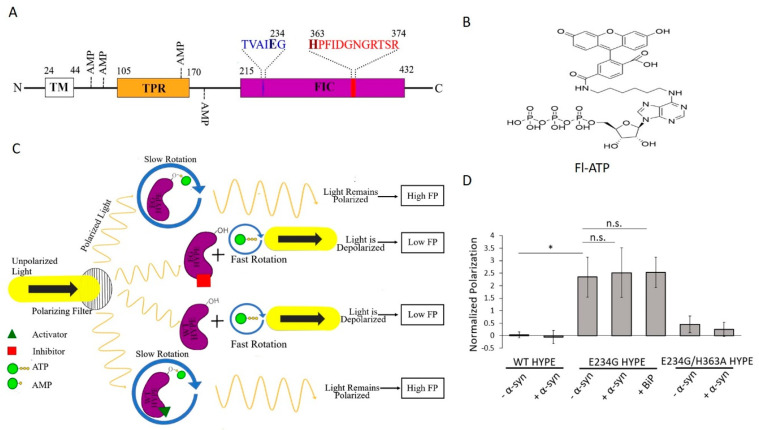
Fluorescence polarization (FP) of Fl-ATP monitors HYPE-mediated AMPylation. (**A**) Schematic depiction of the protein domain architecture of HYPE showing the TM (transmembrane), TPR (tetratricopeptide), and Fic (filamentation induced by cAMP) domains. AMP denotes sites of autoAMPylation. The glutamate-containing inhibitory motif and histidine-containing Fic motif are shown in blue and red, respectively. (**B**) Structure of Fl-ATP (N6-(6-amino)hexyl-ATP-5-carboxyl-fluorescein). (**C**) FP AMPylation assay design. When unattached to wild-type (WT) HYPE or compound-inhibited E234G HYPE (purple), Fl-ATP undergoes rapid rotation to depolarize plane-polarized light, and a basal FP signal is generated. Attachment of Fl-AMP via autoAMPylation (E234G HYPE or compound-activated WT HYPE) allows light to remain polarized for a high-FP signal. (**D**) Proof-of-concept of FP AMPylation assay comparing WT, E234G, or E234G/H363A versions of HYPE_103-456_ (Δ102 HYPE) with or without 10X molar excess of substrates (BiP or α-syn). All samples were normalized to buffer controls. Data are represented as the mean +/− SEM of four replicates. Unpaired *t*-tests were performed: * = *p* < 0.05.

**Figure 2 ijms-21-07128-f002:**
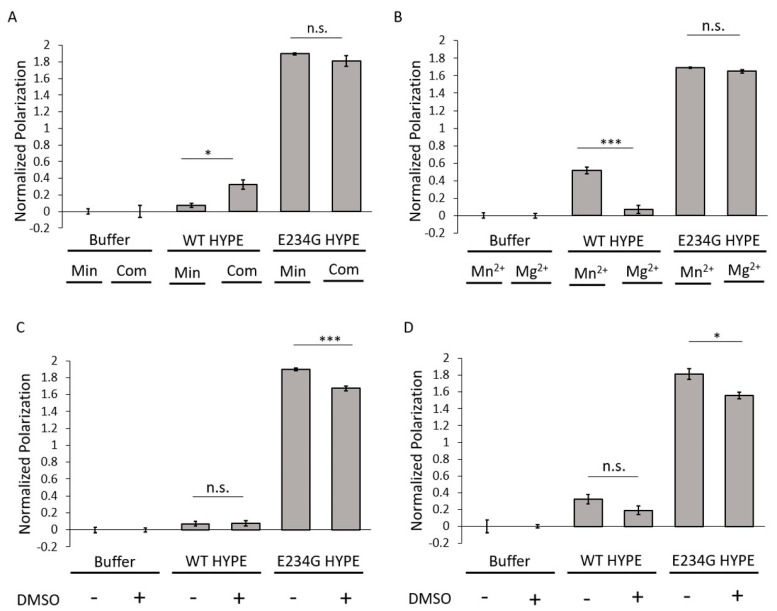
AMPylation buffer assessment. (**A**) Comparison of Δ102 WT and E234G HYPE autoAMPylation in complete (Com) or minimal (Min) buffer (see Methods). (**B**) Comparison of Δ102 WT and E234G HYPE autoAMPylation with buffer containing Mn^2+^ or Mg^2+^. (**C**) Comparison of Δ102 WT or E234G HYPE autoAMPylation with or without 1% DMSO in a minimal buffer background. (**D**) Comparison of Δ102 WT or E234G HYPE autoAMPylation with or without 1% DMSO in a complete buffer background. All samples were normalized to buffer controls. Data represented as the mean +/– SEM of four replicates. Unpaired *t*-tests were performed: * = *p* < 0.05; *** = *p* < 0.001.

**Figure 3 ijms-21-07128-f003:**
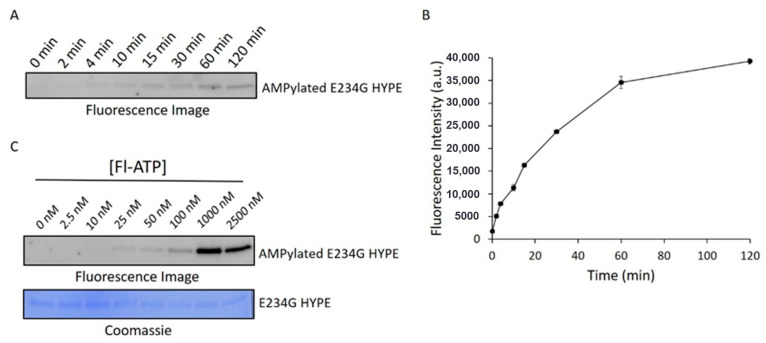
Assessment of AMPylation kinetic parameters. (**A**) Representative fluorescence image of Δ102 E234G HYPE autoAMPylation time-course. (**B**) Quantification of two independent experiments as in (**A**). Data represented as the mean +/− SEM. (**C**) Fluorescence image (top) of Δ102 E234G HYPE autoAMPylation at various Fl-ATP concentrations and corresponding Coomassie image (bottom).

**Figure 4 ijms-21-07128-f004:**
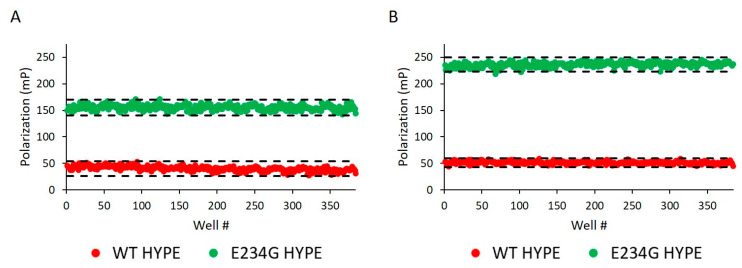
Assay reproducibility assessment. (**A**) 200 nM Δ102 WT and E234G HYPE autoAMPylation reactions were measured for FP on separate 384-well microplates. Z’ = 0.749 and S/B = 3.875. Each dot represents a single AMPylation reaction in a separate well. Black dash lines represent three standard deviations plus or minus the control means. (**B**) As in (**A**) but the mean of two independent 400 nM Δ102 HYPE plates. Mean Z’ = 0.884 and mean S/B = 4.598.

**Figure 5 ijms-21-07128-f005:**
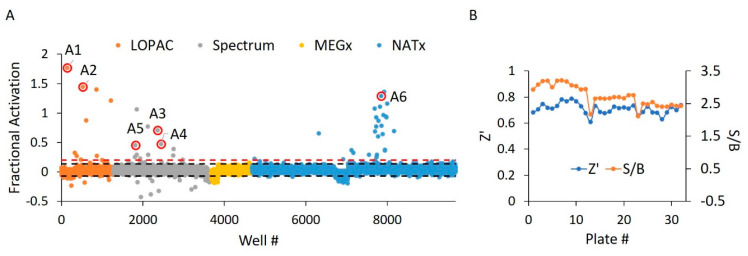
WT HYPE activator high-throughput screen. (**A**) Δ102 WT HYPE FP autoAMPylation reaction incubated with compounds from either LOPAC, Spectrum, MEGx, or NATx libraries. All samples were normalized to Δ102 WT and E234G HYPE controls on the same plate in DMSO-containing buffer. Each dot represents a different compound incubated with a single AMPylation reaction in a separate well. Black dashes represent +/− 3 SD from the mean. Red dashes are the 20% hit definition, with all dots above this threshold being hit compounds. Dots with red circles show hits selected for follow-up, orthogonal validation. (**B**) Plate-to-plate variability in the positive (E234G HYPE) and negative (WT HYPE) DMSO controls from WT HYPE high-throughput screening (HTS). Left and right y-axes are for Z’ and S/B values, respectively. Each dot represents the calculated Z’ or S/B values from 24 E234G HYPE positive controls and 24 WT HYPE negative controls within a single plate.

**Figure 6 ijms-21-07128-f006:**
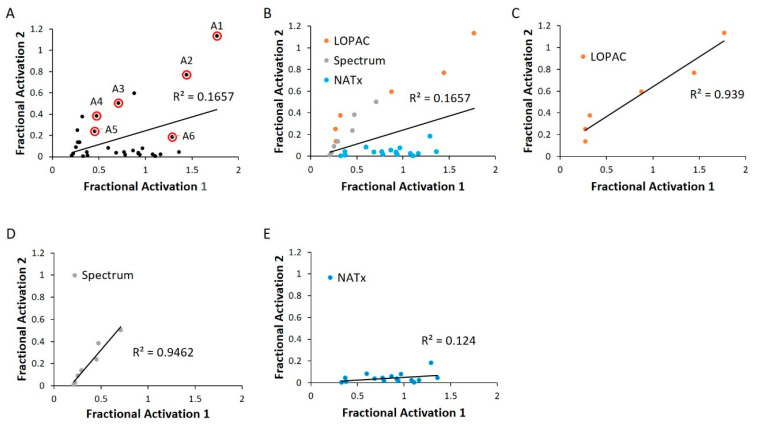
WT HYPE activator validation. (**A**) Comparison between initial HTS compound activity (*x*-axis) and second-pass validation (*y*-axis) with the same FP AMPylation assay. All validation compounds were assayed on the same plate and normalized to internal positive (E234G HYPE) and negative (WT HYPE) DMSO controls. Each dot represents the same compound incubated in two independent AMPylation reactions. Dots with red circles show hits selected for follow-up, orthogonal validation. (**B**–**E**) Data reprocessed from (**A**) to show compound validation among the various libraries.

**Figure 7 ijms-21-07128-f007:**
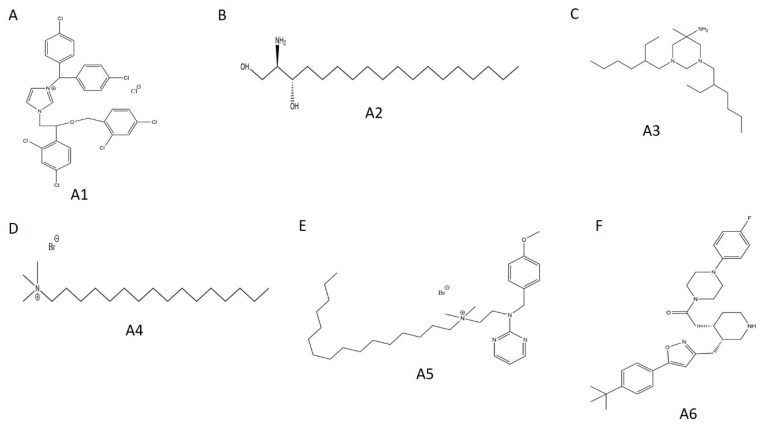
Chemical structures of selected WT HYPE activators. (**A**) Compound **A1**. (**B**) Compound **A2**. (**C**) Compound **A3**. (**D**) Compound **A4**. (**E**) Compound **A5**. (**F**) Compound **A6**.

**Figure 8 ijms-21-07128-f008:**
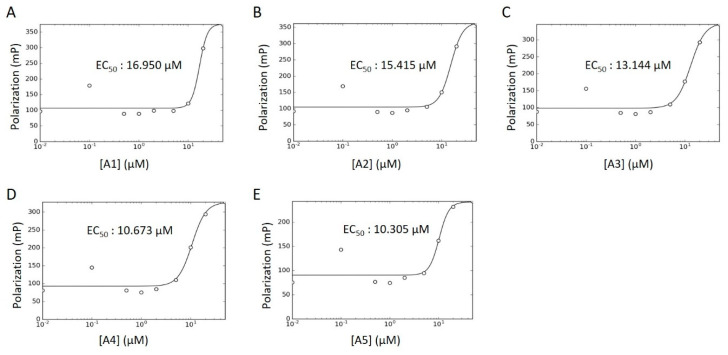
WT HYPE activator concentration–response curve. (**A**–**E**) FP plots from Δ102 WT HYPE AMPylation reactions incubated with 0 to 200 μM of DMSO-dissolved activators. All data were fitted to Equation (5) (see Methods) to determine EC_50_ values. All reactions were done in detergent-less buffer.

**Figure 9 ijms-21-07128-f009:**
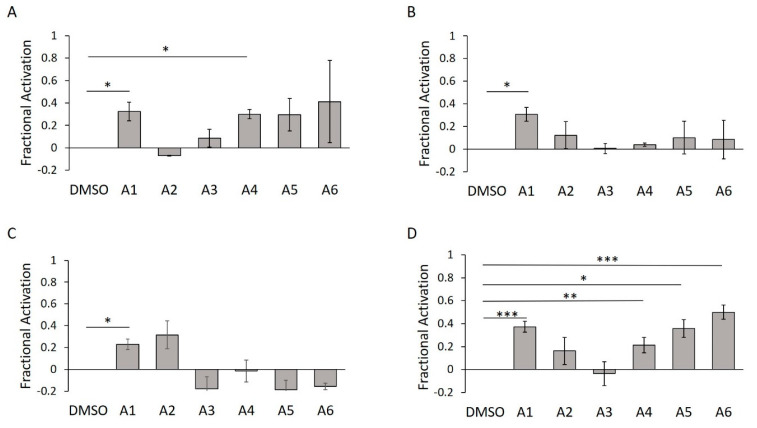
WT HYPE activator validation using in-gel fluorescence. (**A**) Quantification of Δ102 WT HYPE in-gel fluorescence autoAMPylation reaction in minimal buffer without detergent. (**B**) As in (**A**), but with 0.1% Triton X-100. (**C**) As in (**A**), but with Δ45 WT HYPE. (**D**) As in (**A**), but with Δ45 WT HYPE and 0.1% Triton X-100. Quantified data represented as the mean +/− SEM of three independent experiments. Unpaired *t*-tests were performed: * = *p* < 0.05; ** = *p* < 0.01; *** = *p* < 0.001.

**Figure 10 ijms-21-07128-f010:**
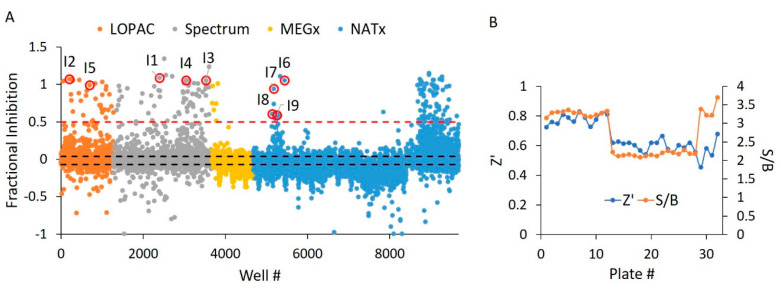
E234G HYPE inhibitor high-throughput screen. (**A**) Δ102 E234G HYPE FP autoAMPylation reaction incubated with compounds from either LOPAC, Spectrum, MEGx, or NATx libraries. All samples were normalized to Δ102 WT and E234G HYPE controls on the same plate in DMSO-containing buffer. Black dashes represent +/− 3 SD from the mean. Red dashes are the 50% hit definition, with all dots above this threshold being hit compounds. Each dot represents a different compound incubated with a single AMPylation reaction in a separate well. Dots with red circles show hits selected for follow-up, orthogonal validation. (**B**) Plate-to-plate variability shows no difference between the positive (Δ102 E234G HYPE) and negative (Δ102 WT HYPE) DMSO controls. Left and right y-axes are for Z’ and S/B values, respectively. Each dot represents the calculated Z’ or S/B values from 24 E234G HYPE positive controls and 24 WT HYPE negative controls within a single plate.

**Figure 11 ijms-21-07128-f011:**
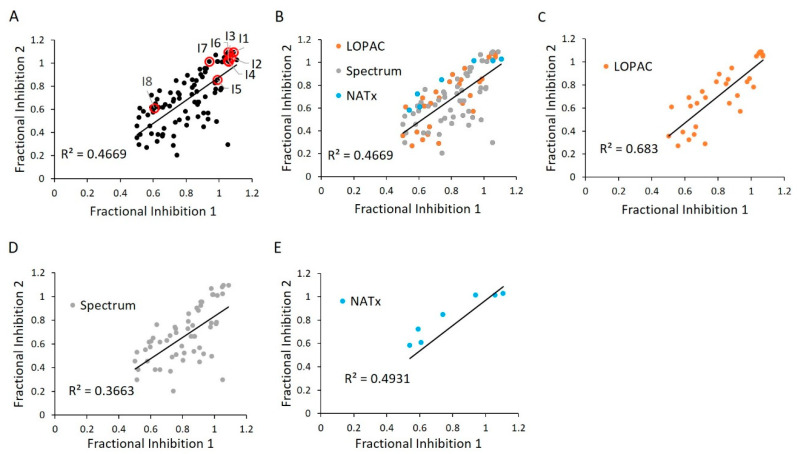
E234G HYPE inhibitor validation. (**A**) Comparison between initial HTS compound activity (*x*-axis) and second-pass validation (*y*-axis) with the same FP AMPylation assay. All validation compounds were assayed on the same plate and normalized to internal positive (E234G HYPE) and negative (WT HYPE) DMSO controls. Each dot represents the same compound incubated in two independent AMPylation reactions. Dots with red circles show hits selected for follow-up, orthogonal validation. (**B**–**E**) Data reprocessed from (**A**) to show compound validation among the various libraries.

**Figure 12 ijms-21-07128-f012:**
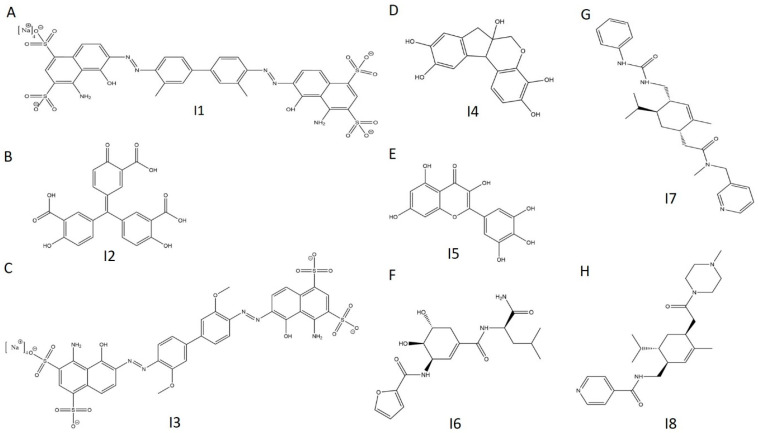
Chemical structures of selected E234G HYPE inhibitors. (**A**) Compound **I1**. (**B**) Compound **I2**. (**C**) Compound **I3**. (**D**) Compound **I4**. (**E**) Compound **I5**. (**F**) Compound **I6**. (**G**) Compound **I7**. (**H**) Compound **I8**.

**Figure 13 ijms-21-07128-f013:**
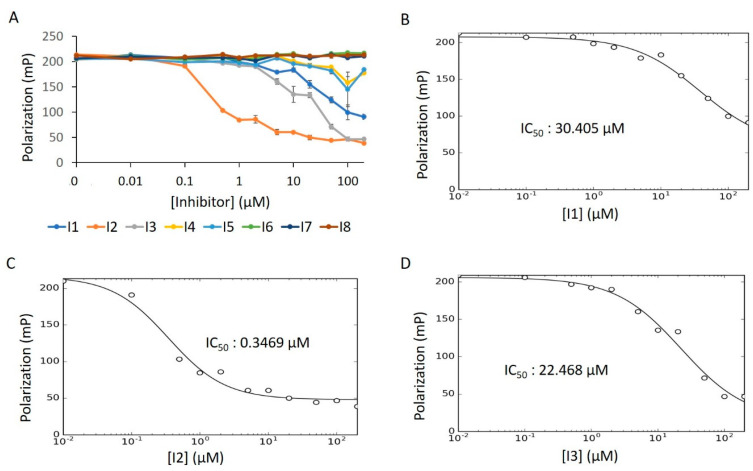
E234G HYPE inhibitor concentration–response curve. (**A**) FP plots from Δ102 E234G HYPE AMPylation reactions incubated with 0 to 200 μM of DMSO-dissolved inhibitors. (**B**–**D**) Data from (**A**) fitted to Equation (6) (see Methods) to determine IC_50_ values. All reactions were done in buffer containing 0.1% Triton X-100.

**Figure 14 ijms-21-07128-f014:**
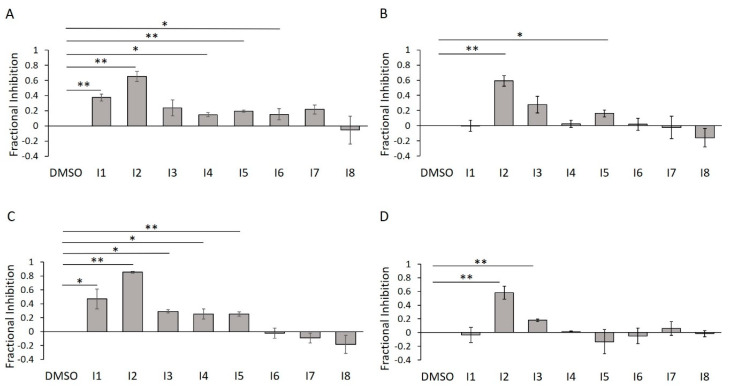
E234G HYPE inhibitor validation using in-gel fluorescence. (**A**) Quantification of Δ102 E234G HYPE in-gel fluorescence autoAMPylation reaction in minimal buffer without detergent. (**B**) As in (**A**), but with 0.1% Triton X-100. (**C**) As in (**A**), but with Δ45 E234G HYPE. (**D**) As in (**A**), but with Δ45 E234G HYPE and 0.1% Triton X-100. Quantified data represented as the mean +/− SEM of three independent experiments. Unpaired *t*-tests were performed: * = *p* < 0.05; ** = *p* < 0.01.

**Table 1 ijms-21-07128-t001:** Compound library information.

Library Name	Company	Number of Compounds	Chemical Properties
LOPAC	Sigma-Aldrich	1280	Bioactive, FDA-approved
Spectrum	Microsource	2400	Bioactive and natural compounds
NATx	AnalytiCon Discovery	5000	Synthetics derived from natural products
MEGx	AnalytiCon Discovery	1000	Natural products
Total	-	9680	-

**Table 2 ijms-21-07128-t002:** WT HYPE activator hit information.

Compound ID	Chemical Formula	MW (Da.)	Chemical Name	Library	% Act. 1	% Act. 2	% Mean Act.
**A1**	C_31_H_23_Cl_7_N_2_O	688	Calmidazolium	LOPAC	177	114	146
**A2**	C_18_H_39_NO_2_	302	DL-erythro-dihydrosphingosine	LOPAC	144	77	111
**A3**	C_21_H_45_N_3_	340	Hexetidine	Spectrum	71	50	61
**A4**	C_19_H_42_BrN	364	Cetyltrimethylammonium Bromide	Spectrum	47	38	43
**A5**	C_32_H_55_BrN_4_O	592	Thonzonium Bromide	Spectrum	45	24	35
**A6**	C_31_H_39_FN_4_O_2_	519	NAT14–350426	NATx	129	18	74

**Table 3 ijms-21-07128-t003:** E234G HYPE inhibitor hit information.

Compound ID	Chemical Formula	MW (Da.)	Chemical Name	Library	% Inh. 1	% Inh. 2	% Mean Inh.
I1	C_34_H_24_N_6_Na_4_O_14_S_4_	961	Evan’s Blue	Spectrum	109	109	109
I2	C_22_H_14_O_9_	422	Aurintricarboxylic Acid	LOPAC	107	106	107
I3	C_34_H_24_N_6_Na_4_O_16_S_4_	993	Chicago Sky Blue	Spectrum	106	109	108
I4	C_16_H_14_O_6_	302	Hematoxylin	Spectrum	105	30	68
I5	C_15_H_10_O_8_	318	Myricetin	LOPAC	99	86	93
I6	C_18_H_25_N_3_O_6_	379	NAT2–252122	NATx	106	102	104
I7	C_27_H_36_N_4_O_2_	449	NAT28–408090	NATx	94	102	98
I8	C_24_H_36_N_4_O_2_	413	NAT28–405040	NATx	61	61	61

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
