# Peer review of "A Fluorescence Polarization-Based High-Throughput Screen to Identify the First Small-Molecule Modulators of the Human Adenylyltransferase HYPE/FICD"

_ijms, 2020, doi:10.3390/ijms21197128_

Round 1

Reviewer 1 Report

he article entitled "A fluorescence polarization-based high-throughput screen to identify the first small molecule modulators of the human adenylyltransferase HYPE/FICD" submitted for publication in IJMS by Camara et al. presents and validates a Dual High throughput screening for inhibition and active HYPE-mediated AMPylation.

The article is well written and well structured. The subject is well introduced and the references cited are appropriate. Figures are detailed and clear and are presented with high quality.

The authors describe in the detail the assay and they demonstrate it to be amenable to automate HTS for diverse chemical libraries. The robustenss and scalability of the test were also demonstrated. Different chemical libraries are tested. A reliable pipeline for hit validation is presented. The level of details presented is quite comple and the article is easy to follow.

In general a very well designed study and a very professional paper that is a pleasure to read.

Author Response

We are very grateful to Reviewer 1 for appreciating our hard work. The team of authors on this manuscript include a mid-career graduate student, three undergraduates, and a high school student. Reviewer 1’s positive feedback has been a great source of encouragement for these budding scientists. As a mentor, such positive reinforcement for my students from Reviewer 1 means a lot – thank you, again!

Reviewer 2 Report

The manuscript is well-conceived and written. I'd like to suggest only the following revisions:

  • Figures concerning the 3D structure of the biological target could be inserted, in order to help the reader in the comprehension of the paper.
  • For the same reason, The Authors could revise their manuscript by molecular docking studies of the proposed derivatives.
  • For these compounds, the Authors could discuss their bioavailability and other pharmacokinetic properties, such as blood brain barrier permeation, water solubility, binding to plasmatic properties. These data could be investigated in silico.
  • For these compounds toxicity could be also predicted, such as hERG inhibitor ability.
  • Quality of figure 12 is ppor. please, revise structures.
  • a brief scheme reporting the common chemical features exhibited by the selected compounds could be added. Please, compare these information with those by docking.

Author Response

Response to Reviewer 2: Reviewer’s comments are indicated in italic. Our responses are in normal.

We thank Reviewer 2 for their helpful suggestions, all of which we have incorporated into our revised manuscript.

Figures concerning the 3D structure of the biological target could be inserted, in order to help the reader in the comprehension of the paper.

For the same reason, The Authors could revise their manuscript by molecular docking studies of the proposed derivatives.

We agree that this study could be improved by in silico docking of our top hit compounds to structures of HYPE. We therefore added a molecular docking analysis of HYPE in complex with our most bioactive inhibitor—I2 (aurintricarboxylic acid). This revision is now presented in Figure S8, and described in the main article in Lines 426-430, and in the corresponding Material and Methods section 4.6.

For Figure S8, the structure of WT HYPE (PDB 4u04) was selected instead of E234G-HYPE because 1) there are no available apo (without nucleotide) structures of E234G-HYPE; 2) in all screening assays our enzymes were preincubated with compounds before the addition of Fl-ATP, thereby making WT-HYPE a more representative structure; and 3) there are no discernible structural differences between WT and E234G HYPE structures (as per Bunney et al., Structure, 2014). Due to software limitations in modeling molecules with multiple heavy atoms (i.e., chlorine), we were unable to dock our most bioactive activator, A1 (calmidazolium) with HYPE. As A1 was our only activator with consistently high activity across all assays, we opted not to assess the docking of other activators.

Our author contributions have also been revised to reflect the new molecular docking work and Figure S8.

For these compounds, the Authors could discuss their bioavailability and other pharmacokinetic properties, such as blood brain barrier permeation, water solubility, binding to plasmatic properties. These data could be investigated in silico.

For these compounds toxicity could be also predicted, such as hERG inhibitor ability.

Our study represents a proof-of-concept for in vitro detection of compound-mediated changes in HYPE-mediated AMPylation, and sets the stage for subsequent expansion of our pilot HTS into larger libraries which will account for compounds with more physiologically relevant properties, such as BBB (blood brain barrier) permeability. Given the lack of bioactivity of most hits in the validation assays, we deemed in-depth analysis of their pharmacokinetic properties meaningless at this stage. However, a statement regarding the adherence of aurintricarboxylic acid to Lipinski’s rule of five was added to the article (Lines 428-429) with appropriate reference. In the case of calmidazolium, its membership in LOPAC1280 (Library of Pharmacologically Active Compounds) is suggestive of its biological activity. We found no published data indicating whether our top hits can permeate the BBB (blood brain barrier). However, as noted in the Introduction, HYPE is implicated in many other disease pathways besides neurodegeneration.

Quality of figure 12 is poor. please, revise structures.

The scheme showing the top inhibitor hits—Figure 12—was revised to improve image quality. The activator structures in Figure 7 were also revised for the same reason.

A brief scheme reporting the common chemical features exhibited by the selected compounds could be added. Please, compare these information with those by docking.

Analysis of the chemical structures of hits yielded no common moieties exceot for the following two groups: 1) ATP analog inhibitors and 2) activators resembling A6 (NAT14-350426). ATP analogs were not advanced for follow-up validation because of their presumed promiscuity towards cellular ATP-binding proteins. A6 and its analogs had low bioactivity in validation assays, obviating schematic representation of their structures.

We again thank the reviewer for their constructive feedback. We believe we have addressed all the reviewer’s concerns in good faith and to the best of our timely ability.

Thank you for your consideration.

Sincerely,

Seema Mattoo